Generation of raptor diversity in Europe: linking speciation with climate changes and the ability to migrate

http://orcid.org/0000-0002-8697-5647 Negro Juan J. 1 negro@ebd.csic.es
Rodríguez-Rodríguez Eduardo J. 2
http://orcid.org/0000-0001-7882-135X Rodríguez Airam 3 4 5
Bildstein Keith 6
1 Estación Biológica de Doñana-CSIC, Department of Evolutionary Ecology , Sevilla , Spain
2 Department of Integrated Sciences, Faculty of Experimental Sciences, Universidad of Huelva , Huelva , Spain
3 Grupo de Ornitología e Historia Natural de las islas Canarias (GOHNIC) , C/La Malecita s/n, Buenavista del Norte, Canary Islands , Spain
4 Terrestrial Ecology Group (TEG-UAM), Department of Ecology, Universidad Autónoma de Madrid , Madrid , Spain
5 Centro de Investigación en Biodiversidad y Cambio Global (CIBC-UAM), Universidad Autónoma de Madrid , Madrid , Spain
6 Blandon, PA , USA
Nazareno Alison
Electronic publication date: 2022 Dec 8
Publication date: 2022
Volume: 10
Electronic Location ID: e14505
Received 2022 Jun 22; Accepted 2022 Nov 13
Copyright: © 2022 Negro et al.
Copyright year: 2022
Copyright holder: Negro et al.
License: This is an open access article distributed under the terms of the Creative Commons Attribution License, which permits unrestricted use, distribution, reproduction and adaptation in any medium and for any purpose provided that it is properly attributed. For attribution, the original author(s), title, publication source (PeerJ) and either DOI or URL of the article must be cited.
License URL: https://creativecommons.org/licenses/by/4.0/

Keywords: Glacial cycles, Species richness, Birds of prey, Speciation, Migration, Interspecific competition

Funding: The authors received no funding for this work.

==============================
Europe holds a rich community of diurnal birds of prey, and the highest proportion of transcontinental migratory raptorial species of any landmass. This study will test the hypotheses that the high diversification of the raptor assemblage in Europe is a recent event, that closely related species sharing the same trophic niches can only coexist in sympatry during the breeding period, when food availability is higher, and finally that migration is a function of size, with the smaller species in every trophic group moving further. A consensus molecular phylogeny for the 38 regular breeding species of raptors in Europe was obtained from BirdTree (www.birdtree.org). For the same species, a trophic niche cluster dendrogram was constructed. Size and migratory strategy were introduced in the resulting phylogeny, where trophic groups were also identified. Multispecific trophic groups tended to be composed of reciprocal sister species of different sizes, while monospecific groups (n = 3) were composed of highly specialized species. Many speciation events took place recently, during the glacial cycles of the Quaternary, and size divergence among competing species may be due to character displacement. Nowadays, the smaller species in every trophic group migrate to sub-Saharan Africa. This investigation illustrates how the rich assemblage of diurnal birds of prey in Europe, more diverse and more migratory than, for instance, the North American assemblage at equivalent latitudes, has emerged recently due to the multiplication of look-alike species with similar trophic ecologies, possibly in climate refugia during cold periods.

Introduction

Raptor migration is a conspicuous phenomenon in many locations globally (Bildstein, 2018). Numerous raptors, 183 out of 313 (58%) species worldwide, abandon their breeding grounds and, often gathering in large flocks, move away at predictable times and places to reach distant areas located sometimes in different continents (Bildstein, Smith & Yosef, 2007; Ferguson-Lees & Christie, 2001). Raptor migration has been extensively monitored to assess long-term population trends and to determine raptor flight behavior, migration corridors, as well as breeding and non-breeding grounds (Bildstein, 2006). However, much remains to be investigated concerning the causes and function of raptor migration, or why a majority of species are migratory but still others remain sedentary. In addition, several works link migratory behavior in birds to increased diversification and speciation (Rolland et al., 2014; Oatley, Simmons & Fuchs, 2015).

Recent molecular advances have shown that there are in fact two distantly related groups of diurnal raptors, convergent in their morphology and general ecology (Jarvis et al., 2014). On one side, are the true falcons and allies, in the Order Falconiformes. On the other side, eagles, hawks, kites, harriers, and vultures constitute the Order Accipitriformes, with the New World vultures (Family Cathartidae or Order Cathartiformes, depending on the source, Del Hoyo, Elliott & Sargatal, 1994; Del Hoyo, 2020) forming a sister group of all remaining Accipitriformes (Jarvis et al., 2014). According to Ericson (2012), the early radiation of Falconiformes took place in South America, whereas Accipitriformes emerged in Africa. Today, these two orders have gained a cosmopolitan distribution and species of the two orders tend to co-occur in most environments. There are many migratory species or populations within both Falconiformes and Accipitriformes (Bildstein, 2006) in both the Nearctic and the Palearctic, implying that long-distance migratory behavior has evolved multiple times and that it is a rather plastic trait (Nagy, Végvári & Varga, 2017).

Europe is the land mass with the highest proportion of migratory species of birds of prey in the world (Bildstein, 2006), and there is also a large absolute number of species considering its relatively small area compared to the remaining of the Palearctic and also the Nearctic. North America, for instance, doubles the surface of Europe and, at similar latitudes and climates, holds a smaller raptor community with fewer long-distance migratory species. Why this is so has never been explained nor explored so far. Also overlooked, even though it should be obvious glancing at any bird guide (e.g., Svensson et al., 1999) or raptor biology book (Ferguson-Lees & Christie, 2001), is the fact that there are numerous groups (mainly duets and trios) of morphologically and ecologically similar raptorial species in the European portion of the Western Palearctic. The species conforming these groups may look so similar in appearance, even if differing in body size, that their field identification at a distance may be challenging (Negro, 1991; Forsman, 2016). We refer to pairs of species such as the Eurasian (Falco tinnunculus) and the lesser kestrels (F. naumanni), the lesser and greater spotted eagles (Clanga pomarina and C. clanga), or trios including, for instance, Montagu’s harrier (Circus pygargus), hen harrier (C. cyaneus) and pallid harrier (C. macrourus).

Morphologically and ecologically similar species living in sympatry are expected to use common resources within a scenario of competition, that may be particularly acute in apical predators preying on animal prey species (Schoener, 1983; Chen, Li & Li, 2022). The wider availability of trophic resources during the reproductive period allows cohabitation of species of different size in the same niches resulting in a denser community packing. However, during winter, larger species may monopolize their respective niches, obligating smaller species to migrate. The role of interspecific competition in the evolution of migration is an old idea in community ecology (e.g., Cox, 1968). How it may operate for raptorial species is still an open question.

Considering evolutionary time, Europe was greatly affected by Pleistocene glaciations in the last 2.5 million years, which had a profound effect on the composition and distribution of genetic lineages of practically all organisms living in this landmass (Drovetski et al., 2018). The European Pleistocene refugial paradigm implies that a large part of Europe was covered by ice on glacial times, forcing temperate and boreal taxa into milder southern refugia. In interglacial times, populations recolonized vacant northern regions, but isolation and demographic reductions in the refugia for thousands of years often had led to behavioral and genetic divergence which impeded genetic admixture of populations originating in different areas. The first reviews of glacial species included poorly dispersive organisms such as mammals, trees or insects (Hewitt, 1996, 2000; Taberlet et al., 1998). Those seminal works, however, lacked examples of birds, by definition organisms with high dispersal capabilities (Drovetski et al., 2018). Concerning raptors, two cases conforming to the described paradigm, and leading to full speciation, have been described: two eagle species (Aquila adalberti and A. heliaca, Ferrer & Negro, 2004), and two kite species (Milvus migrans and M. Milvus, Roques & Negro, 2005). Another example in owls (Pellegrino et al., 2014), involves two different genetic clades within the same taxonomic species (Athene noctua) in Europe. A review of the refugial paradigm for the whole community of diurnal birds of prey is not yet available.

Here we aim to link interspecific competition, raptor migration, and finally diversity and speciation in the European diurnal raptor assemblage, including both Falconiformes and Accipitriformes. Controlling for phylogenetic relatedness and considering some regularities in size differences and migratory behaviour, we will test several hypotheses: (a) the raptor community in Europe has recently experienced a diversification of breeding species, mainly during the Quaternary. This hypothesis would be supported if the split from the common ancestor of sister species tended to take place within the last 2.5 million years according to time-calibrated phylogenies. (b) the coexistence of multiple look-alike species in the same trophic niches is only possible during the breeding season (i.e., spring and summer), when food resources are less limiting in Europe. The immediate prediction is that ecologically similar species should segregate in the winter time, either by changing habitats or food habits, or migrating and vacating the area in which they would compete. (c) The propensity to migrate may be a function of body size, with smaller species in every trophic niche moving further than the larger species, which should be superior competitors (Schoener, 1983). Size-biased migration in winter times is predicted, however, by both the Bergmann’s rule (Bergmann, 1847), and under a scenario of agonistic competition and predation (Schoener, 1983; Hakkarainen & Korpimäki, 1996; Grether et al., 2009).

Methods

Study area and selected species

This study focuses on the 38 diurnal raptor species breeding regularly and historically in continental Europe (see Table 1). We use the term Europe as defined in the seven-continents model (i.e., Europe separated from Asia as a conglomerate of peninsulas delineated by the Mediterranean sea to the South, the Atlantic ocean to the West, the Northern sea in high latitudes, and the Ural mountains, the Caucasus and the Black and Caspian seas to the East) because it suits well the differences in migrating behaviour of birds including raptors in northern and temperate areas (e.g., Herrera, 1978; Newton & Dale, 1996; Newton & Dale, 1997; Bildstein, 2006). Using continental Europe as a biogeographical area allows us to cleanly separate sedentary, migrants within continental Europe, and sub-Saharan bird species migrating through the Mediterranean flyways (i.e., Straits of Gibraltar, Messina and Bosphorus). However, the Western Palearctic, a zoogeographical region widely used in bird studies (e.g., Cramp & Simmons, 1979; Perrins & Snow, 1998), and that fully encompasses Europe, introduces African and Asian raptorial species that would add noise to our analyses. These species, either at the southern limit of the Western Palearctic in the African continent or at the eastern border with Asia, are all rare breeders or vagrants in the area that have never established breeding populations in continental Europe. Raptors that may reach the Western Palearctic but are clearly African taxa include, at least, the lappet-faced vulture (Torgos tracheliotos), the dark chanting goshawk (Melierax metabates), the barbary falcon (Falco pelegrinoides), the sooty falcon (Falco concolor), the tawny eagle (Aquila rapax), and Verreaux’s eagle (A. verreauxi) (Svensson et al., 1999). The steppe eagle (Aquila nipalensis) has not been considered because the majority of its breeding population occurs in Asia.

Table 1 Assemblage of diurnal raptors breeding in continental Europe, ordered according to trophic niche.

Assemblage of diurnal raptors breeding in continental Europe, ordered according to trophic niche (group, first column). Some of the groups (two to four) are monospecific, as they are constituted for highly niche-specific species. Range of body masses for males and females were taken from Ferguson-Lees & Christie (2001), as well as their migratory status (i.e., trans-Saharan migrant or not). IUCN conservation status is also given (IUCN, 2021): LC, least concern; VU, vulnerable; NT, near threatened; EN, endangered.

Group	Species	Body mass male (g)	Body mass female (g)	Transaharian migration	IUCN	
1	Pandion haliaetus	1,120–1,740	1,210–2,050	Yes	LC	
1	Haliaeetus albicilla	4,900–6,000	6,800–9,000	No	LC	
2	Circaetus gallicus	1,200–2,000	1,300–2,300	Yes	LC	
3	Pernis apivorus	440–943	450–1,050	Yes	LC	
4	Circus aeruginosus	405–730	540–960	Yes	LC	
5	Circus macrourus	235–416	402–550	Yes	NT	
5	Circus pygargus	227–305	254–445	Yes	LC	
5	Circus cyaneus	300–400	370–708	No	LC	
6	Accipiter gentilis	517–1,110	820–2,200	No	LC	
6	Accipiter brevipes	135–223	232–275	Yes	LC	
6	Accipiter nisus	105–196	185–350	Yes	LC	
7	Milvus migrans	630–928	750–1,080	Yes	LC	
7	Milvus milvus	757–1,220	960–1,600	No	LC	
8	Buteo buteo	427–1,180	486–1,360	Yes	LC	
8	Buteo lagopus	600–1,130	783–1,660	No	LC	
8	Buteo rufinus	590–1,281	945–1,760	Yes	LC	
9	Hieraaetus pennatus	510–770	840–1,250	Yes	LC	
9	Aquila fasciata	1,560–1,560	2,000–2,500	No	LC	
10	Aquila chrysaetos	2,800–4,600	3,800–6,700	No	LC	
10	Aquila adalberti	2,500–3,500	2,500–3,500	No	VU	
10	Aquila heliaca	2,450–2,720	3,160–4,530	Yes	VU	
11	Clanga clanga	1,700–1,900	1,800–2,500	No/Yes	VU	
11	Clanga pomarina	1,000–1,400	1,300–2,200	Yes	LC	
12	Aegypius monachus	7,000–11,500	7,500–12,500	No	NT	
12	Gyps fulvus	6,200–10,500	6,500–1,100	Yes	LC	
13	Gypaetus barbatus	4,500–7,000	5,600–6,700	No	NT	
13	Neophron percnopterus	1,600–2,400	1,600–2,400	Yes	EN	
14	Falco eleonorae	350–390	340–460	Yes	LC	
14	Falco subbuteo	131–232	141–40	Yes	LC	
15	Falco peregrinus	550–750	740–1,300	No	LC	
15	Falco pelegrinoides	549–610	657–924	No	LC	
15	Falco biarmicus	500–600	700–900	No	LC	
15	Falco rusticolus	800–1,320	1,130–2,100	No	LC	
15	Falco cherrug	730–950	970–100	Yes	EN	
16	Falco naumanni	90–172	128–216	Yes	LC	
16	Falco vespertinus	115–190	130–197	Yes	NT	
16	Falco tinnunculus	136–252	154–314	No	LC	
16	Elanus caeruleus	197–277	219–343	No	LC	

Phylogeny and trophic niche

To determine the genetic relationships among the different species under study, and more specifically to determine sister species, a phylogeny was obtained from BirdTree (Jetz et al., 2012, 2014), a free online tool providing validated phylogenies for any subset of extant bird species (www.birdtree.org, hereafter BirdTree). The tree construction approach of BirdTree combines relaxed clock molecular trees of well-supported avian clades with a fossil calibrated backbone with representatives from each clade. As explained in Rubolini et al. (2015), BirdTree uses as a backbone for phylogenetic reconstruction two major reference works by Hackett et al. (2008) and Ericson et al. (2006). In our case, we downloaded 1,000 trees from the Ericson backbone. A consensus phylogeny was constructed using FigTree v 1.4.4 (Rambaut & Drummond, 2018) after 1,000 permutations. The phylogenetic tree that we used is the least-squares consensus tree calculated from the mean patristic distance matrix of a set of 1,000 probable phylogenies. The consensus tree for our subset of 38 European raptorial bird is not time calibrated. For dating splits of sister species, we used the two most recent time calibrated molecular phylogenies available for raptors worldwide (i.e., Fuchs, Johnson & Mindell, 2015; Mindell, Fuchs & Johnson, 2018).

Addressing migratory behaviour in birds requires not only the consideration of biogeographic aspects, but also a knowledge of diet specializations and prey availability (Nagy & Tökölyi, 2014). A trophic niche cluster dendrogram, based on binary distances, was constructed using a dataset with data of trophic niche variables of each species, including hunting or scavenging behaviour, prey type and size (i.e., small birds, large birds, small mammals, large mammals, arthropods, herps or fishes), scavenging strategy and foraging method (Fig. 1). Data were obtained from two reference books, the Handbook of the Birds of the World (Del Hoyo, Elliott & Sargatal, 1994), and Raptors of the World (Ferguson-Lees & Christie, 2001). These data can be consulted as supplemental material (Table S2). Categorical observations were transformed into numeric ones to build a matrix with presence/absence (i.e., 1 or 0), to calculate the binary distances used for cluster construction. Scavenging type and foraging method were disaggregated to create binary variables for each level. A niche dendrogram was constructed in R environment (R Core Team, 2022) using hclust function (method = complete). The R script is included as supplemental material. Presentation of figures was improved with Inkscape 1.0.1, adding images of raptors and colours (Inkscape Project, 2020). Additionally, in order to validate the trophic groups obtained above, we conducted a K-means algorithm-based cluster analysis (Jain, 2009) using fviz_dist and kmeans functions implemented in R environment (R Core Team, 2022). We selected 16 centres, as with the first cladogram. The R script is given as supplemental material.

Figure 1 Trophic niche cluster dendrogram based on binary distances for the community of European breeding raptors (n = 38 species).

Picture credits: Gypaetus barbatus, Accipiter nisus and Falco eleonorae by M. Cayuela. Circus pygargus by M. A. Rojas. The remaining pictures are by the authors.

Relating body size and mass with migration distance

We compiled the following information of the raptor species: the minimum and maximum value of both total length and wingspan (four variables) from Del Hoyo, Elliott & Sargatal (1994), the minimum and maximum values of length, tail, and tarsus (six variables), as well as ranges of male and female body masses (four mass values in total) from Ferguson-Lees & Christie (2001). To estimate body mass, we averaged the minimum and maximum values for males and females. To estimate bird size, we ran a principal component analysis (PCA) on the above mentioned 14 centered and scaled morphometric variables. The first principal component was used as a body size index. The first principal component retained 90.1% of variation (see PCA details in Table S1). The fourteen morphometric variables showed the same sign for their factor loadings and highly significant correlations to the first principal component (Fig. S1). Factor loadings of the fourteen morphometric variables and importance of the components are given in Table S1. Pairwise correlations of the first principal component (body size index) and seven morphometric variables (i.e., those representing the lowest values of the ranges of each morphometric) are shown in Fig. S2.

Based on results from trophic niche clustering, groupings of species sharing a similar niche were established. These groups were identified on the previously built phylogeny. Using body mass data and migratory strategy (Table 1), we established two categories: smaller raptors inside their groups and larger raptors inside their groups, and estimated the proportion of trans-continental migratory species among large and small species. Frequency of migration for both groups of large and small species were tested using a contingency table. In addition, we plotted the mass of paired migratory and resident species in the foraging and phylogenetic groups that we had previously identified. By adding a diagonal line (y = x), the resulting plot graphically shows in which cases the migratory species was either larger (above the diagonal) or smaller (below the diagonal) than its resident or less-migratory counterpart. We additionally plotted the previously described body size index, because the anatomical design of migratory species is known to be slender, i.e., much longer wings, or different wing loads, compared to non-migratory species.

Distributions ranges of raptor species were downloaded from the IUCN Red List Data (IUCN, 2021) and imported in Qgis (v3.20) (QGIS Development Team, 2021). We deleted passage areas for migrating species and mapped pairs and trios of species according to the analysis of trophic niche.

Results

Foraging groups

We obtained 16 trophic or ecological niche groups (Fig. 1) based on binary distances of behavioural and dietary variables, replicated with the K-means cluster analysis (Fig. S1). The identified niches included three monospecific niches (i.e., the reed-bank specialist Circus aeruginosus, the snake eating Circaetus gallicus and the hymenopteran eating Pernis apivorus), seven groups of two species, four groups of three species, one group of four species, and one group of five species conformed by the bird-eating falcons (Table 1, Fig. S3).

Phylogenetic relationships of European raptors

The resulting phylogeny for the European raptor species matches well with the obtained trophic groups in a majority of cases, except for the Pandion haliaetus/Haliaeetus albicilla and Aquila fasciata/Hieraaetus pennatus species pairs, which are polyphyletic (Fig. 2).

Figure 2 Phylogeny of European raptors.

A phylogeny of European raptors with indication of shared trophic niches and migratory status. Coloured areas on species names represent trophic niches. Red asterisks indicate long-distance migration. Coloured stars join species not phylogenetically directly related but sharing same trophic niche. Numbers 1 to 3 denote species exploiting monospecific niches (1. Circus aeruginosus; 2. Circaetus gallicus; 3. Pernis apivorus). Picture credits: Clanga clanga modified from J. M. Garg (CC BY-SA 3.0). Buteo lagopus cropped from Walter Siegmund (CC BY 2.5). Haliaeetus albicilla cropped from Christoph Müller (CC BY 4.0). Falco biarmicus cropped from Derek Keats (CC BY 2.0). Falco rusticolus from NorthernLight (CC BY-SA 3.0). F. vespertinus and F. cherrug with permission from B. Rodríguez. F. eleonorae and Accipiter gentilis with permission of Manuel Cayuela. Remaining pictures by the authors. License links: https://creativecommons.org/licenses/by-sa/3.0/legalcode; https://creativecommons.org/licenses/by/2.5/legalcode; https://creativecommons.org/licenses/by/4.0/legalcode; https://creativecommons.org/licenses/by/2.0/legalcod3.

Implications of size and migration strategy

When plotting the body masses for paired species derived from the foraging groups (Fig. 3A), the values fall below the diagonal of the graph (x axis, mass of less migratory species, y axis, mass of more migratory species in the pair). This demonstrates that the fully or more migratory species are systematically smaller than the less migratory or resident species. Essentially the same results are obtained when instead of plotting body masses, body size indices were plotted (Fig. 3B). Larger species are the less migratory ones in the species pairs. The distribution ranges for a selection of the species pairs that we considered in the above plots have been mapped in Fig. 4. The less-migratory species remain within continental Europe (although it has to be noted that some species may have resident populations in Africa such as the peregrine falcon and the Eurasian kestrel), whereas the fully migratory species have non-overlapping breeding (in Europe) and non-breeding ranges (in Africa).

Figure 3 Pairwise comparisons of body mass and body size.

Pairwise comparisons of body mass (A) and body size (B) of raptor species derived from the foraging groups in Fig. 1. Y-axis represents the migratory species in the pair, while the X-axis the non-migratory species (resident). Dashed lines indicate the diagonal (y = x). Lowercase letters refer to species pairs: (a) Gypaetus barbatus—Neophron percnopterus, (b) Aegypius monachus—Gyps fulvus, (c) Clanga clanga—Clanga pomarina, (d) Aquila chrysaetos—Aquila heliaca, (e) Aquila fasciata—Hieraetus pennatus, (f) Circus cyaneus—Circus macrourus, (g) Circus cyaneus—Circus pygargus, (h) Accipiter gentilis—Accipiter brevipes, (i) Accipiter gentilis—Accipiter nisus, (j) Buteo rufinus—Buteo buteo, (k) Buteo rufinus—Buteo lagopus, (l) Milvus milvus—Milvus migrans, (m) Falco tinnunculus—Falco naumanni, (n) Falco tinnunculus—Falco vespertinus, (o) Falco peregrinus—Falco eleonorae, (p) Falco peregrinus—Falco subbuteo, (q) Falco rusticolus—Falco columbarius, (r) Falco rusticolus—Falco cherrug.

Figure 4 Distribution maps.

Year-round distribution maps, with pairs or trios of species resulting from the trophic-niche cluster analysis. Ranges of larger-in-size species are displayed in purple, while ranges of the smaller and typically intercontinental migratory species are displayed in orange (and green in case of trios). Distribution ranges were taken from the IUCN Red List Data (IUCN, 2021).

A majority of the smaller species inside trophic groups (three species in monospecific groups were excluded) reach Africa during migration (88% of cases, n = 17). On the contrary, only 22% (four out of 18) of the large species inside trophic groups perform a trans-Saharan migration (i.e., Clanga clanga, Falco eleonorae, Circus macrourus and Buteo rufinus). However, these “large” species, except F. eleonorae, which overwinters in Madagascar, perform a much shorter migration than their smaller counterparts in their respective groups (see Fig. 4). The difference in migratory behaviour among larger and smaller species is highly significant ( X2 = 12,8075, P = 0.000345, Yates correction applied).

Discussion

A comparison of raptors communities in Europe and North America

The diversity of species of any given taxa in a land mass typically depends on its own evolutionary history, the interaction with other species in the area, and the past and present physical patterns of the environment (e.g., Hansen & Urban, 1992; Mönkkönen, 1994). Our study was prompted by the fact that the number of breeding species of diurnal raptors in continental Europe (38 species in a land-mass of 10,180,000 km2) is higher than the number of breeding species of raptors in Canada and continental US combined (30 species in about 19,000,000 km2, including the three endemic New World vultures), and also by the disproportionately high number of transcontinental migratory species in Europe (Bildstein, 2006, see below). The latitudinal range is much wider in the American side and includes large areas below 35° North latitude in the southern states of the US that are not represented in continental Europe. However, this does not bring over a significant number of tropical species that do inhabit Mexico and Central America (the number of raptorial species including the latter area rises the total to 69 species, Clark & Schmitt, 2017).

Seven species are shared between Europe and North America: five Holarctic raptor species, the gyrfalcon, the merlin, the golden eagle, the rough-legged hawk, and the northern goshawk, plus two cosmopolitan species, the peregrine falcon and the osprey, which hold breeding populations in all continents except Antarctica (Ferguson-Lees & Christie, 2001). Leaving aside these shared species, mainly from northern or mountainous regions, there is a marked difference among Europe and Canada-US combined in the number and proportion of regular transcontinental migratory species, which is much higher in Europe (22 species crossing the Mediterranean Sea to Africa in the non-breeding season) than in the above mentioned area of North America, where only five species make the bulk of the migrants moving down to South America (i.e., Ictinia mississippiensis, Buteo swainsoni, B. platypterus, Cathartes aura and Pandion haliaetus) (Bildstein, 2006).

The high diversity of European raptors

Our results show that there is a duplicity, or even multiplicity of species in the same trophic and ecological niches in Europe. In many cases the species involved are so similar that identification at a distance may be difficult (Forsman, 2016). This similarity may be due to shared ancestry in congeneric species, such as in the duet conformed by C. pomarina and C. clanga, or in the case of M. milvus and M. migrans (Mindell, Fuchs & Johnson, 2018), but it may be due to convergence in more distantly related species such as A. fasciata and H. pennatus. Duplicated species are not apparent in North America, where the two more diverse genera (Buteo and Falco), have rather distinct species in terms of size and ecology. That Europe has been a recent hotspot for raptor speciation is also supported by the pattern of endemism (Weir & Schluter, 2004) in the group: six species are only or mainly distributed in Europe (i.e., red kite, honey buzzard, lesser spotted eagle, Spanish imperial eagle, Montagu’s harrier and Eleonora’s falcon). Five other species breed in Europe and more into the East, but may have a Western Palearctic origin: lesser kestrel (with the older fossil found in the Iberian Peninsula, Negro et al., 2020), red-footed falcon (sister species of the Amur falcon breeding in eastern Asia), common buzzard (the nominal subspecies is sedentary across Europe, and only the Eastern subspecies B. b. vulpinus migrates to Africa), and finally the levant sparrowhawk, and the western marsh harrier.

Timing of the splits from the common ancestor

According to the time-calibrated phylogenies of falcons (Falconiformes) on the one hand (Fuchs, Johnson & Mindell, 2015) and of all other birds of prey (eagles, hawks, vultures and kites in Accipitriformes) on the other hand (Mindell, Fuchs & Johnson, 2018), six of our species groupings (both in the phylogeny and in the trophic cladogram) originated in the Quaternary (i.e., the last 2.5 million years) as sister species. These groupings are the following: (1) The red and the black kite. (2) The two spotted eagles, lesser and greater. (3) The two imperial eagles (A. heliaca and A. adalberti), that we have matched ecologically to the most distantly related A. chrysaetos. (4) The three European buzzards: common, rough legged and long-legged. (5) The two late-breeding falcons, dependent on the return migration of passerines for reproduction: hobby and Eleonora’s falcon. (6) The large European falcons: lanner, saker and gyrfalcon. Apart from this, we may add that the two colonial and tree-nesting falcons of temperate Eurasia, i.e., the red-footed falcon in the Western Palearctic and Amur falcon in the Eastern Palearctic, which are also sister species splitting from a common ancestor whose range may have spanned the temperate forests of the whole of Eurasia.

The remaining groupings may have formed earlier in time, again according to Fuchs, Johnson & Mindell (2015) and Mindell, Fuchs & Johnson (2018): (1) the Eurasian kestrel and the lesser kestrel emerged as species in the kestrel radiation of the Pliocene, about 4–5 million years (Negro et al., 2020). (2) The harriers (hen, Montagu´s and pallid) may have radiated about 5 million years ago. (3) The hawks of the genus Accipiter radiated much earlier, and the common ancestor of the northern goshawk and both the European sparrowhawk and levant sparrowhawk possibly evolved 10–15 million years ago in the Miocene. (4) The two small and ventrally white eagles, Bonelli’s and booted, emerged from a common ancestor living in the Pliocene 4 million years ago. (5) The two large European vultures, griffon and cinereous, shared a common ancestor about 9 million years ago. (6) Coalescence time for Bearded and Egyptian vultures, which are sister groups to each other, is about 12 million years.

Taken together, our results support the notion that the current raptor assemblage in Europe experienced a recent enrichment, and may be considered a hotspot accounting for its area and geographic location, and particularly by comparison with the smaller raptor community in much larger North America. This does not mirror the situation for the whole of the avifaunas, as there are about 556 bird species in Europe (Keller et al., 2020), compared to 740 in North America to the north of Mexico (Cornell lab of Ornithology, https://www.notesfromtheroad.com/roam/how-many-birds-north-america.html).

The role of Pleistocene glacial cycles

As we recounted in the introduction, one mechanism that may explain the generation of raptor diversity in Europe is the Pleistocene refugial paradigm, i.e., the existence of climate-driven vicariance events during the glacial cycles of the Pleistocene (Voelker, 2010; Drovetski et al., 2018). None-the-less, climatic changes resulting in speciation may have occurred earlier during the Pliocene or in the Miocene. It has been suggested, for instance, that the genus Falco radiated during the Miocene several million years ago, coincidental with a period of increased aridity favouring grasslands (Zachos et al., 2001; Fuchs, Johnson & Mindell, 2015). Glacial cycles have been invoked as drivers of speciation for numerous bird taxa in both the Old and the New World (Bermingham et al., 1992; Weir & Schluter, 2004; Lovette, 2005). Such a model was in fact invoked to explain the emergence as separate species of the Eastern Imperial Eagle and the Spanish imperial eagle (Ferrer & Negro, 2004). The latter is the species in the genus Aquila with the smallest distribution range, as it is endemic to the Iberian peninsula.

Lacking enough dated fossils, the recent avian speciation during the Pleistocene glacial cycles has remained controversial for some groups (Lovette, 2005; Zink, Klicka & Barber, 2004; Wang et al., 2018). However, recent advances in molecular techniques, such as whole-genome sequencing, clearly support that many bird species suffered dramatic expansions and contractions during the Quaternary coincidental with climate changes (Nadachowska-Brzyska et al., 2015). These population changes may have prompted speciation.

The generation of sister species of birds of prey may be explained by vicariance events. The quantitative difference among Europe and North America, comparing their respective raptor communities, may have to do with the different effects of the glaciation in the two land masses, and the alignment of the major mountain barriers that may have acted as geographical barriers, with a north-south component in the Nearctic and east-west in the Palearctic (Albach, Schönswetter & Tribsch, 2006). In North America, sister species tend to have east-west distributions (e.g., Lovette, 2005). In Europe, the existence of several southern peninsulas and islands offered different refugia and thus more possibilities for population fragmentation and isolation (Covas & Blondel, 1998). After ice cover retreated during the interglacials, isolated populations might have expanded and established contact but, in some cases, without genetic admixture.

Agonistic competition and character displacement

After species separated in reproductive isolation, a second mechanism may have started to play a role in amplifying specific differences once species reunited after glaciations. Closely related species of predators coexisting in sympatry would compete dearly in the harsh winter season (see, e.g., Chen, Li & Li, 2022). Competition would be resolved by the less competitive species moving away and thus migrating to a different continent. This may also be seen as several concurrent cases of character displacement (Brown & Wilson, 1956), with competing species diverging in size precisely to avert or reduce competition (Dayan & Simberloff, 1998). The mechanism of character displacement has been invoked to account for morphological differences of closely related species, often congeneric, living in sympatry (Yousefi et al., 2017). Some classical examples include Darwin’s finches (Grant & Grant, 2006), and therefore character displacement is thought to play a role in speciation and the emergence of adaptive radiations.

Also of interest to account for the high diversity of European birds of prey may be the fact that the assemblage of European carnivores (Mammalia) is much smaller, with just 22 species, than the North American assemblage, with 45 species (Burgin et al., 2020). After all, mammalian and avian predators often prey on the same resources, including, for instance, rodents, lagomorphs and carrion. Lessened potential competition with mammalian carnivores may have facilitated the diversification of European breeding raptors. None-the-less, agonistic competition (Grether et al., 2009) with other predators alone is not the only possible explanation for the smaller raptors performing longer migrations, as Bergmann’s rule (Bergmann, 1847), for instance, would also predict larger bird species having a thermoregulatory advantage in northern climates compared to smaller species during the cooler non-breeding season (Salewski & Watt, 2017). As additional support to our hypotheses, the three New World vultures of North America conform to the size and migration pattern that we propose, with the largest species, the California Condor, being sedentary, and the much smaller species turkey and black vultures being migratory. The same can be said of the two NorthAmerican congeneric hawks, with the smaller species, the sharp-shinned hawk (Accipiter striatus) typically being more migratory than the largest Cooper’s Hawk (A. cooperii) (Ferguson-Lees & Christie, 2001).

An extraordinary case of rapid increase of size in the absence of competitors, therefore linking body size to competition in the raptor group, is provided by two extinct birds of prey in New Zealand (Knapp et al., 2019): the largest known eagle in the world, Haast’s eagle Hieraaetus moorei, and the largest known harrier in the world, Eyles’ harrier Circus teauteensis. Both derived from Australian vagrants, the much smaller Hieraaetus morphnoides and Circus assimilis, respectively. These two concurrent cases of island gigantism occurred in the early Pleistocene when the clearing of closed forests due to climate changes allowed for successful colonization of Australian raptor immigrants hunting in open woodlands and grasslands (Knapp et al., 2019). Long-distance dispersal and migration has also invoked in the speciation of other Circus species (Oatley, Simmons & Fuchs, 2015). The fact that body size is in turn related to migration propensity and ability is also supported at the intraspecific level. Raptors tend to present reversed sexual dimorphism, as females are larger than males (Newton, 1979). Numerous raptor species show sex-biased migration timing and non-breeding distributions, although it is true that there is variation in which sex migrates first (both out-bound and in-bound) or overwinters in more northerly locations (see Bildstein, 2006).

Final remarks

We wish to call attention to the fact that a majority of the raptorial species in Europe have an unfavorable conservation status (Burfield, 2008, see also Table 1) and some, as with the Spanish imperial eagle and the bearded vulture, were on the brink of extinction in the late 20th century (Bustamante, 1998; Martínez-Cruz, Godoy & Negro, 2004). The system we have described here seems to be unique to Europe and has resulted in a hotspot for raptors, many of which are migratory and face conservation problems both inside and outside Europe. The current scenario of climate change adds uncertainty to population persistence (Donázar et al., 2016; Rodríguez-Rodríguez et al., 2020), while some trans-Saharan migratory species are already changing their behaviour and becoming non-migratory in southern Europe. This includes at least the Egyptian vulture (Di Vittorio et al., 2016), lesser kestrel (Negro, De la Riva & Bustamante, 1991), booted eagle (Mellone et al., 2013) and short-toed eagle (Martínez & Sánchez-Zapata, 1999). In addition, African species are entering Europe from the south, as it possibly happened with the black-shouldered kite in historical times (Balbontín et al., 2008; Rivera et al., 2022), and it is just starting with the Ruppell’s vulture (Ramírez et al., 2011). If interspecific competition and character displacement were key factors to explain diversity, distribution and migratory behaviour in the European raptor community, as our results suggest, the rapid contemporary changes in climate and bird’s ecology will favour some species at the expense of others, generating new conservation challenges.

Supplemental Information

Supplemental Information 1 Species involved and raw data.

Click here for additional data file.

We thank M. A. Rojas, M. Cayuela, B. Rodríguez, J. M. Irastorza, J. M. Sayago and A. Kovacs for sharing pictures included in Figs. 1 and 2, and S3. We also thank all the photographers who have shared pictures under Creative Commons licenses.

Additional Information and Declarations

Competing Interests

Author Contributions

Data Availability

The authors declare that they have no competing interests.

Juan J. Negro conceived and designed the experiments, performed the experiments, analyzed the data, authored or reviewed drafts of the article, and approved the final draft.

Eduardo J. Rodríguez-Rodríguez performed the experiments, analyzed the data, prepared figures and/or tables, authored or reviewed drafts of the article, and approved the final draft.

Airam Rodríguez performed the experiments, analyzed the data, prepared figures and/or tables, authored or reviewed drafts of the article, and approved the final draft.

Keith Bildstein performed the experiments, authored or reviewed drafts of the article, and approved the final draft.

The following information was supplied regarding data availability:

The data and code used for preparing Fig. 1 (Trophic niche cluster dendrogram) are available in the Supplemental File.

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
