# Peer review of "Generation of raptor diversity in Europe: linking speciation with climate changes and the ability to migrate"

_PeerJ, doi:10.7717/peerj.14505_

## Round 0.1 · original submission · Major Revisions

I sent the manuscript out for review to determine whether the referees would identify the merits of the study that would justify publication for PeerJ. Although the reviewers recognized the merits, they mention limitations, raising some misgivings about the way the manuscript has been written up and analyzed. They mentioned that (1) Introduction should be better structured, including elements that can sustain the hypothesis presented, (2) the methods section needs to be clarified, informing how the analyses was performed, and (3) a deeper discussion regarding the central topic of this work is required. I hope that you will find all advice helpful when revising the manuscript.

Reviewer 1 ·

Basic reporting

This is an interesting and well-written manuscript about the ecological and evolutionary factors behind the high diversity of raptors that co-occur in Europe. The dataset and methods seem appropriate for the study goals, and the figures look great -- I particularly love the inclusion of pictures of these beautiful birds alongside the data. The hypothesis that smaller species have evolved migratory syndromes to reduce competition with larger relatives is a fascinating one.

My major comments and suggestions can be divided into five parts:

1. Provide additional background on the Introduction.
2. Provide more structure on the hypotheses tested (also in the Intro).
3. Provide additional details on some of the Methods.
4. Potentially add one figure to the Results to show patterns of divergence times obtained from previous investigations.
5. Include an additional discussion on the issue of how trophic and body size divergence might be related to character displacement driven by competition.

Specific comments:

1. Introduction: Background. I appreciate the Introduction's conciseness (it is well-written despite being very short), but I have the impression that a few concepts could be introduced more deeply. This manuscript integrates many interesting aspects of the ecology and evolutionary history of raptors in Europe, and I commend the authors on that integrative approach. Still, I missed some background on each of these aspects. For instance, one component of this investigation appears to be how climatic fluctuations during the Quaternary have contributed to raptor speciation. Still, the Introduction provides little to no background on the extensive literature about refugial speciation in Europe (or elsewhere). Moreover, an essential aspect of the paper is divergence in trophic niche allowing for the coexistence of many raptor species in Europe. Still, there's virtually no background information on the role of ecological divergence and associated character displacement in speciation and species packing within ecological assemblages. Expanding the Introduction a bit (including additional sentences and perhaps one or two paragraphs on the issues I mentioned) would provide additional conceptual background, and better lay the foundation for this investigation, particularly for unfamiliar readers.

2. Introduction: Hypotheses. I noted that this manuscript aims to test two distinct hypotheses: 1) An evolutionary hypothesis related to the rapid and recent speciation of European raptors, and 2) another hypothesis related to the ecological coexistence ("community packing") of many species of raptors through trophic niche differences. I can even note a third implicit hypothesis: that the propensity to migrate is a function of body size, with smaller species being more migratory than larger ones. Note that each of these hypotheses outlines different predictions, and testing them requires distinct approaches. It could be helpful if the authors more explicitly outlined that multiple (interrelated) hypotheses are being tested instead of one (as stated at the end of the Introduction). In stating these hypotheses, the authors could then outline some predictions and the approaches used to test each. As of now, many different ideas are packed into a single hypothesis and associated goal. However, this investigation comprises many interesting tests that could be introduced more structured and explicitly.

3. Methods, line 123. Please explicitly state why you needed a phylogenetic tree: "To determine THIS, we obtained a phylogenetic tree...". It is implicit that you need a tree to establish the timing of species divergences and determine pairs of sister species to compare trophic niche overlap patterns. However, having that stated explicitly in the Methods would be helpful.

4. Related to the above, please provide more information about the tree used. What is the ultimate source of the tree hosted at BirdTree? Is that coming from one or several trees? How were these trees generated? Are they molecular phylogenies? What exactly is BirdTree? Is this a time-calibrated phylogeny? How was it calibrated? Please provide a brief general description of the tree used and how it was derived.

5. Methods, line 125. Please provide information on how the trophic niche dendrogram was constructed. What were the sources of the trophic variables used? Are they coming from published studies? Online compendia of bird biology? Personal observations? Please clarify and provide additional detail on how the data was gathered.

6. Similarly, how was the trophic data transformed and processed? Did you use PCA analyses? How did you convert the categorical observations into numeric variables to be amenable to estimating Euclidean distances? Perhaps using a presence/absence (i.e., zeros and ones) matrix? Please clarify and provide additional detail on how the data were processed and analyzed. Note that you did a good job describing how you processed the morphometric data in the paragraph below -- that's precisely the type of information and structure I think is missing for the trophic data.

7. Still related to the above, what R packages were used for each analysis? Would it be possible to provide the corresponding R scripts as supplementary materials?

8. Results, line 170. It seems the trophic/ecological clusters were established more or less arbitrarily based on the structure of the dendrogram. Of course, the dendrogram is hierarchical and could be split at different levels to compose other partitions of subclusters. The dendrogram appears to have been divided based on the species' biology instead of using some unsupervised clustering algorithm. I don't think this is a problem at all. Still, perhaps it would be helpful to provide a statement explaining the reasoning behind how the dendrogram was partitioned into trophic/ecological clusters.

9. Results, line 202. The discussion of divergent times in this section cites two studies that appear to have performed divergence time estimation (Fuchs et al. 2015 and Mindell et al. 2018). Still, from the Material and Methods, I had the impression that these dates would have come from the tree(s) obtained in BirdTree. Reading the Results, it appears that the Methods did not mention how divergence times were obtained. I would find it essential to modify the manuscript to describe in the Methods how divergence times were obtained, the trees and studies used, and a brief description of how these trees were built (e.g., molecular data) and dated (e.g., fossil calibrations).

10. Related to the comment above, I think it would be important to provide a figure presenting the calibrated phylogenies obtained from those previous studies. It doesn't seem enough to describe the topologies obtained from earlier investigations in your Results section, particularly if no corresponding figures are provided. In fact, this section feels more like Discussion material.

11. I found the Discussion pretty interesting. Still, it focused on the (refugial) biogeography of speciation and migration patterns in European raptors compared to other bird groups or raptors in different regions of the world (e.g., the Americas). However, I missed a deeper discussion of a central topic of this study: divergence in trophic niches and potentially body sizes as a result of character displacement to reduce ecological overlap and competition (mentioned only briefly in the last paragraph of the manuscript). Please consider adding additional sentences or a section focused on these topics. The focus on historical biogeography in the Discussion seemed only tangentially related to the goals and content of this investigation, which appeared to be more focused on the issue of phenotypic and ecological overlap between co-occurring species. It is worth noting that climate refugia have also been implied to explain speciation in North America and are not exclusive to Europe. As such, it is unclear how such refugia might explain the high diversity of European raptors relative to North American ones, something the authors discuss extensively in the text.

I hope the authors find my comments pertinent and useful. Thanks for considering my suggestions and for the opportunity to read this cool manuscript.

Experimental design

Nothing to add -- please refer to my basic reporting above.

Validity of the findings

Nothing to add -- please refer to my basic reporting above.

Additional comments

Nothing to add -- please refer to my basic reporting above.

Reviewer 2 ·

Basic reporting

General comments
The manuscript by Negro et al. is very well written and brings interesting aspects of the evolution and diversity of diurnal raptorial birds in Europe. Using the information on species size and ecological traits such as trophic niche and migratory strategy in conjunction with a time-calibrated molecular phylogeny, the authors support the hypothesis that most speciation events occurred during Quaternary glacial cycles and that smaller raptor species perform long-distance migrations in comparison to larger species. Most of my comments are to improve the manuscript and the overall understanding of the issues addressed:

Experimental design

Lines 201-226. The entire session "Timing of the splits from the common ancestor" should be moved to the manuscript discussion. The Results session should be reserved for results obtained by the authors during the development of the study, not comparisons and discussions of results obtained in other studies - this belongs in the Discussion session. The authors mainly used the results of the time-calibrated phylogenies of falcons published in other studies but did not produce results from their data.

Validity of the findings

Lines 94-95. I suggest that the authors include more information about the reasoning behind the hypothesis regarding the timing of diversification. Is there a specific reason to test the hypothesis of diversification during the Quaternary? Some background in the introduction could help readers to better understand the hypothesis proposed in the study.

Lines 228 – 251. The first two paragraphs of the discussion are comparisons between the diversity of diurnal raptors in North America and Europe. These comparisons are outside the study objectives, according to the last paragraph of the Introduction. The authors need to include this comparison among the study objectives for better understanding and engagement of the readers.

Lines 217. The authors should provide more background on the "Northern origin" hypothesis in the introduction of the manuscript. Mentioned in the discussion as it is now, it adds an unanticipated complexity to the flow of the text.

Additional comments

Figure 2. Since the authors are discussing divergence times between species pairs, it would be interesting to use a phylogeny with the "average time" of separation between the taxa represented at the nodes. For better visualization, it would also be helpful if the authors included a bar indicating the geological eras (Pleistocene, Pliocene, etc.) below the phylogeny.

·

Basic reporting

the article follows all those sections but the hypotheses could be formulated a bit clearer (end of introduction) as to match the very well structured results section.

Experimental design

research questions could be formulated clearer (see above comment also). there are a few small details missing from the methods section, which I noted in the pdf directly. (example: the fourteen mass variables used for the size indices are not clear from the text)

Validity of the findings

the statistics are rather simple but straightforward and match the research questions well. But the actual results could be discussed more directly, as currently the discussions jumps right into the "big picture" whereas I suggest to first discuss your tested findings here and then move out to more speculated aspects of the study subject

Additional comments

the paper reads very well and interesting, almost like a textbook but in a good and refreshing way. the study questions are interesting and worth a publication

---

## Round 0.2 · Minor Revisions

Thank you very much to send an improved version of your manuscript. However, some points still need to be clarified. Please address all suggestions carefully. I hope that you will find all advice helpful when revising the manuscript.

Reviewer 1 ·

Basic reporting

I commend the authors for their work in this resubmission. This manuscript version is much improved, and I enjoyed reading it. Fascinating system and set of hypotheses! I thank the authors for considering my previous suggestions, and I have only a few minor requests for clarification that will hopefully improve this manuscript further, as I detail below.

Line 29. Please replace "moving further" with "traveling higher distances" (if this is what you meant here).

Line 14. Thank you for improving your explanation of how you obtained a phylogenetic tree. There is one aspect that still needs to be clarified, though. The authors mentioned that "BirdTree combines relaxed clock molecular trees of well-supported avian clades with a fossil calibrated backbone with representatives from each clade". However, they then state, "The consensus tree for our subset of 38 European raptorial birds is not time-calibrated". I have two questions:

1) Aren't the first and second statements contradictory regarding whether the tree is calibrated? I apologize if I'm missing something obvious here.

2) If you performed time calibration, what was the procedure? Penalized likelihood (e.g., using TreePL), perhaps? Please clarify and adjust as needed.

Line 265. What about Accipiter hawks in North America? Cooper's and Sharp-shinned hawks come to mind -- the second being smaller and more migratory, I believe (but please verify). Is it interesting to mention this example?

Line 275. I understand that this was a suggestion by another reviewer, but I'm afraid I have to disagree that the divergence time section should be presented in the Discussion. I understand that you obtained and manipulated phylogenetic information and even performed analyses to re-calibrate the tree (it seems). Therefore, I'd consider this to be Results material, and it definitely reads as such. Of course, it is up to the authors to decide whether to present this section in the Discussion or Results. My goal with this comment is to state that I'd support your initial approach.

Line 323. This paragraph proposes that pairs of sister species have diverged in isolation in different glacial refugia. However, how do you connect this model with the hypothesis of the evolution of character displacement in size? Has size divergence participated in the speciation process or happened afterward to allow species co-existence? Would the histories of speciation and character evolution be different in the case of congeners versus non-congeners that belong to the same trophic group ? I'm having difficulty connecting the different hypotheses tested in this paper into a single evolutionary scenario. Please clarify.

I hope the authors find these comments pertinent, fair, and helpful. Thanks for considering my suggestions and the opportunity to read this exciting contribution.

Experimental design

All good -- much improved version. Please refer to my detailed comments in Basic Reporting, above.

Validity of the findings

All good -- much improved version. Please refer to my detailed comments in Basic Reporting, above.

Additional comments

All good -- much improved version. Please refer to my detailed comments in Basic Reporting, above.

Reviewer 2 ·

Basic reporting

The authors have clearly and diligently answered all the issues raised. Thus, I endorse the article for publication in the PeerJ.

Experimental design

NA

Validity of the findings

NA

Additional comments

NA

---

## Round 0.3 · accepted · Accept

After a careful reading of the revised version of this manuscript, I would like to express my appreciation to the authors for their very good job in answering all the questions raised and I am happy to accept this manuscript in its current form.